# A Collaborative Merging Method for Connected and Automated Vehicle Platoons in a Freeway Merging Area with Considerations for Safety and Efficiency

**DOI:** 10.3390/s23094401

**Published:** 2023-04-30

**Authors:** Huan Gao, Yanqing Cen, Bo Liu, Xianghui Song, Hongben Liu, Jia Liu

**Affiliations:** 1Research Institute of Highway Ministry of Transport, Beijing 100088, China; gaoh0916@163.com (H.G.); boliu2015@163.com (B.L.); sxh@itsc.cn (X.S.); lhb_its@163.com (H.L.); liu.jia@rioh.cn (J.L.); 2Department of Automation, Tsinghua University, Beijing 100083, China

**Keywords:** connected and automated vehicle (CAV), freeway ramp, freeway merging area, mixed traffic flow, collaborative merging strategy, road safety and efficiency, TERCRI

## Abstract

To solve the problems of congestion and accident risk when multiple vehicles merge into the merging area of a freeway, a platoon split collaborative merging (PSCM) method was proposed for an on-ramp connected and automated vehicle (CAV) platoon under a mixed traffic environment composed of human-driving vehicles (HDV) and CAVs. The PSCM method mainly includes two parts: merging vehicle motion control and merging effect evaluation. Firstly, the collision avoidance constraints of merging vehicles were analyzed, and on this basis, a following–merging motion rule was proposed. Then, considering the feasibility of and constraints on the stability of traffic flow during merging, a performance measurement function with safety and merging efficiency as optimization objectives was established to screen for the optimal splitting strategy. Simulation experiments under traffic demand of 1500 pcu/h/lane and CAV ratios of 30%, 50%, and 70% were conducted respectively. It was shown that under the 50% CAV ratio, the average travel time of the on-ramp CAV platoon was reduced by 50.7% under the optimal platoon split strategy compared with the no-split control strategy. In addition, the average travel time of main road vehicles was reduced by 27.9%. Thus, the proposed PSCM method is suitable for the merging control of on-ramp CAV platoons under the condition of heavy main road traffic demand.

## 1. Introduction

Due to the inherent spatial interference and velocity differences between on-ramp vehicles and main road vehicles, the merging area is a major bottleneck restricting the safe and efficient operation of the freeway. According to the “Global status report on road safety” released by the World Health Organization in 2018, about 1.35 million people die from road traffic every year, which is an average of 3700 deaths from car accidents every day, and one person loses his or her life on the road every 24 s [1]. In addition, relevant studies have found that on some metropolitan highways, most congestion is irregular congestion caused by abnormal events. This not only reduces the passing efficiency of vehicles, but also leads to a large number of casualties and property losses. For example, if three of the four lanes of traffic are temporarily closed during an unusual event on a congested road, the combined damage in one minute can be as high as USD 2000 [2].

Especially when the on-ramp traffic demand is high, exceeding the traffic capacity of the merging area, rear-end collisions, side collisions, and other traffic accidents and congestion often occur. In the traditional intelligent traffic control system, the following two methods are mainly used to relieve congestion in the merging areas of a freeway: (i) on-ramp flow control [3,4,5]; and (ii) variable speed limit control [6,7,8]. Both of the above two control methods are elements of nodal control, which is essentially a delay or slowdown of the conflicts. They do not fundamentally improve the vehicle merging mode; it is difficult to actually eliminate the merging conflicts, and the effect on improving the safety and coordination of the vehicle merging process is limited.

The appearance of CAVs changes the situation in which the vehicle relies on the driver for information acquisition, driving decisions and driving operations. CAVs equipped with multiple sensors, such as vision sensors and lidar, can better complete driving tasks in various meteorological conditions and road scenarios [9]. With the help of multi-sensor fusion perception, CAVs can perceive the position, velocity and acceleration information of surrounding vehicles within a certain space range, and this information can be used as the basis for driving decisions. Considering that the velocity and acceleration states cannot be obtained directly, many scholars chose to design robust estimation methods to obtain these states indirectly [10,11]. L. Xiong et al. proposed an automatic estimation method for the sideslip angle and attitude of a vehicle body based on inertial measurement units (IMUs) and aided by low sampling rate Global Navigation Satellite System (GNSS) velocity and position measurements based on a parallel adaptive Kalman filter, which identifies small speed and sideslip errors. However, due to the limitations of vehicle computing power and the accuracy and perception range of vehicle sensors, it is difficult to cope with complex traffic scenes only by depending on a CAV’s own ability. Based on the above reasons, considering the collaboration between CAVs and roadside units (RSUs) provides a possible solution [12,13]. RSUs with certain sensing, computing and communication capabilities are set up at specific locations. The CAVs can receive large-scale and processed traffic information from the RSUs through vehicle-to-X (V2X) communication while passing through. Therefore, most current studies consider the implementation of elaborate controls of freeway merging areas based on V2X technology.

Therefore, on the basis of V2X technology, the control mode based on CAV collaborative merging provides a new idea for solving the problem of multi-vehicle merging in the merging area of a freeway, and this has become a research hotspot. According to the existing research, the core idea of the collaborative merging control method is to coordinate the order and speed of the main road and ramp vehicles through the merge point based on the target lane of the main road and on-ramp traffic flow information, so as to avoid collisions during merging. For the method of determination of merging order, the existing research can be basically divided into two categories: (i) first-in, first-out (FIFO) merging order; and (ii) dynamic optimization merging order.

On the one hand, the control method based on FIFO merging order is usually combined with the virtual vehicle method. By mapping the location of on-ramp vehicles on the main road, so that the main road vehicles and on-ramp vehicles are located in the same lane space, and based on the idea of FIFO to determine the merging order, then the longitudinal control of vehicles can achieve the smooth merging of on-ramp vehicles [14,15,16,17,18]. Aiming at the low utilization efficiency of the upstream and downstream ends of a passenger drop-off area under the principle of FIFO, Yang et al. [18] developed a microscopic simulation model in-house. In the strategy, taxis are prohibited from overtaking each other while dropping off patrons. Further experiments showed that an increase in taxi outflow by more than 25% could be achieved by requiring that taxis discharge patrons when forced by downstream queues to stop a prescribed distance in advance of a desired drop-off area.

For example, Rios-Torres et al. [19] defined a circular region with a certain radius and center of confluence as the specific region and the sequence of vehicles entering the specific region as the confluence sequence of vehicles, and then conducted vehicle trajectory planning, modeling the trajectory planning into an optimal control problem by aiming to optimize vehicle energy consumption. The simulation results show that this method can achieve safe vehicle confluence and reduce vehicle energy consumption. Ding et al. [20] optimized the FIFO confluence order and proposed four rules for changing the confluence order of vehicles, allowing vehicles running in the same lane to form a platoon and pass through the conflict area. The simulation results show that the proposed method achieves a good balance between traffic efficiency and computational cost. Based on the FIFO principle, a novel hybrid system [21] composed of discrete cooperative systems was proposed that can carry out path planning and motion control for each automatic vehicle. This method can eliminate confluence conflicts, reduce vehicle delays and improve traffic efficiency in the merging area to a certain extent. However, there are some problems, such as relatively limited control means and unreasonable confluence order when traffic demand is high.

On the other hand, a control method based on dynamic optimization merging order generally aims to find the main road gaps that meet certain conditions, optimize and solve the merging order of on-ramp vehicles and main road vehicles through various algorithms, and carry out path planning for them [22]. In this method, different vehicle driving modes in the connected environment can be introduced, and vehicles on the main road and on-ramp are regarded as single vehicles, showing that on the main road, the on-ramp vehicles operate as a single vehicle or a platoon. Pei et al. [23] proposed a global optimal sequential solution strategy based on dynamic programming. Heejin Ahn [24] proposed a two-mixed integer linear programming problem formulated for simplified vehicle dynamics to prevent side collisions among vehicles at an intersection. A similar approach was adopted by Li [25] in a study to represent heterogeneous vehicle movements and dynamic signal timing plans. Further, a spanning tree and pruning rule were also proposed to search for the globally optimal passing order [26]. In recent years, on the basis of collaborative control of single on-ramp vehicles and main road vehicles, relevant studies [27,28,29] put forward merging strategies that considered the condition of CAV formation on the main road. In order to reduce the computational complexity of enumerating confluence sequences, Xu [30] proposed a confluence sequence optimization method based on vehicle formation, which divided several vehicles whose time headway was less than a certain threshold into a group. However, the effect of this method on improving traffic efficiency in a confluence area was not analyzed. Similarly, in some studies, conflicts between different vehicles were calculated off-line, which greatly reduced the amount of computation [27]. The key problem for control methods based on dynamic optimization merging order lies in the determination of merging order. A common solution is ergodic search optimization, that is, the optimization objective function values of all possible merging schemes are calculated, so as to select the global optimal scheme. However, in complex scenarios, the feasibility and timeliness of the merging scheme are greatly reduced due to the sharp increase in calculation cost.

To sum up, this study proposed a collaborative merging method for a CAV platoon in a freeway merging area with considerations for efficiency and safety. The on-ramp CAV platoon was divided into several sub-platoons, and a collaborative merging driving strategy was designed for each sub-platoon. Due to the fixed sequence of vehicles in a given sub-platoon, the calculation is simplified to a certain extent, and the level of vehicle coordination is improved. The advantages of this method mainly include the following two aspects:(i).According to the actual distribution of the main road gaps, an appropriate split strategy for the ramp CAV platoon is selected, so that the CAV platoon can complete the merging process safely and efficiently in the form of several sub-platoons. It can make full use of the gap space of the main road under different traffic conditions, especially with heavy traffic, while retaining the advantage of CAV platoon driving, improving the safety and efficiency of ramp vehicles, and reducing interference with the normal flow of the main road vehicles.(ii).The PSCM method proposed in this study does not require the active coordination of main road vehicles in terms of velocity, that is, there is no mandatory requirement for the ratio of CAVs in the main road traffic flow. Therefore, the PSCM method is suitable for the scenario of a highway confluence area where the intellectualization degree of the CAV needs to be improved and the main road penetration is not high.

The remainder of this paper is organized as follows: Section 2 describes the methods of the algorithm proposed in the manuscript. Section 3 designs a simulation experiment to demonstrate the validation of the PSCM method. Section 4 presents a numerical analysis of the experimental results, followed by the study conclusions and a vision for future work in Section 5.

## 2. Methods

### 2.1. An Overview

In this study, a platoon split collaborative merging (PSCM) method was designed for a CAV platoon to be merged in a merging area, considering the impact on traffic flow safety and efficiency. The merging platoon is divided into several sub-platoons, each sub-platoon is assigned an accepted gap on the main road, and the merging vehicle motion control is also designed. And the flow chart of the proposed PSCM method is shown in Figure 1 below:

In this scenario, CAVs exit from the on-ramp in the form of a platoon and flow into the main road through an acceleration lane. The main road vehicles are mixed traffic flow with HDVs and CAVs. Intelligent roadside units are installed upstream of the on-ramp, the main road section and the acceleration lane, respectively, are able to collect vehicle velocity, acceleration, position and other information, and have the function of data communication with CAVs. HDVs are driven entirely on the basis of the driver’s perception and judgment. Meanwhile, a plane rectangular coordinate system is established for the above scene: the positive direction of the x axis is the driving direction of the vehicle, and the origin point O is set at the end of the nose. The schematic of the merging area is shown in Figure 2 below:

In combination with the performance parameters of vehicles in the intelligent platoon, vehicle information within a certain range of the target lane of the main road is obtained based on vehicle–road cooperation, and the strategy of the platoon split is formulated in advance to assist the intelligent platoon in performing the platoon split in the acceleration lane, so that each sub-platoon after splitting can make full use of the gaps in the main road and dynamically merge into the traffic flow, ensuring the efficiency and safety of platoon merging and, as far as possible, reducing the impact on the main road traffic flow.

It is worth mentioning that the following two assumptions were made in this research scenario: (i) vehicle–vehicle communication and vehicle–road communication are reliable with negligible delay; (ii) the stop–start process of the vehicle is not considered.

In this study, the vehicles on the main road run at a constant velocity $v_{main}=80\;\mathrm{km/h}$ in the confluence area, and there is no lane change behavior during the driving process. According to the actual speed limit on Chinese freeways, the speed limit on the outermost lane of the main road is generally 60 km/h, with six lanes or more in both directions. Although 60 km/h is a speed limit considered for freight vehicles, all vehicles in this study are passenger cars. Therefore, the speed limit of the outermost lane should be appropriately raised to 80 km/h. The vehicles to be merged are intelligent vehicles with network connection functions and running at uniform velocity in the single lane on-ramp. It should be noted here that this study focuses on the formulation of a platoon split strategy before ramp vehicle merging and does not involve subsequent merging driving strategies for the time being. The length of the acceleration lane is $L_{acc}$.

In this study, it is necessary to obtain the driving status information on the main road traffic flow so as to analyze the merging conditions of vehicles and formulate the on-ramp vehicle splitting scheme. Among them, the gap for on-ramp vehicles merging into the main road, that is, the spacing in front of each car, is particularly important; the size of the gap has a direct impact on the size of the platoon. However, due to acceleration lane restrictions, on-ramp vehicles may only merge into gaps located within a certain space. Therefore, it was necessary to define the spatial range of main road clearance that can be considered and focus on analysis of the main road clearance in the information acquisition area for this research.

Lower boundary of the main road gap:(1)$$x_{1}=\frac{2\cdot v_{main}\left(v_{main}-v_{ramp}\right)}{a}$$

Upper boundary of the main road gap:(2)$$x_{2}=\frac{v_{main}\cdot L_{acc}}{v_{ramp}}$$

The considered range of the main road gap is:(3)$$\left[x_{O}-\frac{v_{main}\cdot L_{acc}}{v_{ramp}},x_{D}-\frac{2\cdot v_{main}\left(v_{main}-v_{ramp}\right)}{a}\right]$$

### 2.2. Merging Vehicle Motion Control

On the basis of determining the information acquisition area of the main road, we consider splitting the intelligent platoon that is joined by the ramp and designate a main gap that can be joined to each sub-platoon after splitting. Obviously, there is an upper limit for the number of vehicles that can be accommodated inside a certain main road clearance, so, as a constraint for the allocation of the size of the sub-platoon, it is necessary to determine the maximum number of vehicles that can be accommodated in that clearance space to ensure the feasibility and rationality of the split strategy.

#### 2.2.1. Collision Avoidance Conditions

First of all, to meet safe collision avoidance requirements for the platoon, including platoon vehicle collision avoidance and the process of merging with the gap before and after car collision avoidance, in this study, considering that all vehicles on the ramp are intelligent vehicles with network connection functions, the spacing between the front and back of the vehicles in the formation is the safety spacing that can be calculated. Since the vehicles in the platoon follow the CACC model, where $t_{c}=0.6\;s$, that is, the time headway between two adjacent vehicles is 0.6 s, and the initial velocity of the platoon is 60 km/h, the spacing between the front and back of the vehicles in the formation is 60/3.6×0.6=10 m. Collisions between the vehicles with front and rear vehicles in the formation in the process of lane changes are not considered. The collision avoidance conditions for cars before and after the gap during lane changes are analyzed. The collision avoidance diagrams of the ramp vehicle and the gap ahead and behind vehicle are shown in Figure 3 and Figure 4 respectively as follows:

The minimum initial distance between the merging vehicle ${\overset{⌢}{C}}_{i}$ and the vehicle in front of the target lane ${\bar{C}}_{j}$ without collision during the merging process is:(4)$$minS_{\overset{⌢}{i},\bar{j}}\left(0\right)=max\left\{\left[{\overset{⌢}{v}}_{i}\left(0\right)-{\bar{v}}_{{}_{j}}\left(0\right)\right]\cdot t+\int_{0}^{t}\int_{0}^{\tau }\left({\overset{⌢}{a}}_{i}-{\bar{a}}_{j}\right)dtd\tau +W\cdot sin\left(\theta \left(t\right)\right)\right\}+S_{0},t\in \left[T_{cri},T_{fin}\right]$$
where $S_{0}$ is the minimum safe distance between the two adjacent vehicles, which is related to the vehicle following model. The values of $S_{0}$ are 30, 16 and 10 $\left(m\right)$ for the IDM, ACC and CACC models, respectively; $T_{cri}$ represents the critical collision time during lane changing, while $T_{fin}$ represents the time when the vehicle completes the lane change and the longitudinal velocity adjustment in the main road.

Similarly, the minimum initial distance between the merging vehicle ${\overset{⌢}{C}}_{i}$ and the vehicle after the target lane ${\bar{C}}_{j+1}$ without collision during the merging process is:(5)$$minS_{\overset{⌢}{i},\bar{j}+1}\left(0\right)=max\left\{\left[{\overset{⌢}{v}}_{\bar{j}+1}\left(0\right)-{\bar{v}}_{{}_{\overset{⌢}{i}}}\left(0\right)\right]\cdot t+\int_{0}^{t}\int_{0}^{\tau }\left({\overset{⌢}{a}}_{\bar{j}+1}-{\bar{a}}_{\overset{⌢}{i}}\right)dtd\tau +L\cdot cos\left(\theta \left(t\right)\right)\right\}+S_{0},t\in \left[T_{cri},T_{fin}\right]$$

Assuming that $minS_{\overset{⌢}{i},\bar{j}}\left(0\right)=S_{1}$, $minS_{\overset{⌢}{i},\bar{j}+1}\left(0\right)=S_{2}$, the merging vehicle does not collide with the vehicles before and after the clearance, and the constraint conditions for safe merging are:(6)$$\left\{\begin{array}{l} S_{1}-S_{0}>0\\ S_{2}-S_{0}>0 \end{array}\right.$$

That is, when the upper condition is established, a sufficiently safe distance can be guaranteed to achieve collision avoidance when ramp vehicles merge.

#### 2.2.2. Following–Merging Rule

By calculating the minimum safe distance between the merging vehicle and the target lane gap before and after the above, the safety requirements for the merging vehicle are ensured. On this basis, for this study, the position of the leading vehicle in the sub-platoon after it completes the merging maneuver should be as close as possible to the leading vehicle in the gap, so as to make full use of space resources and enable the following vehicle in the same sub-platoon to complete the merging maneuver. Therefore, we proposed a following–merging rule.

In the following–merging rule, after the front vehicle in the sub-platoon merges into the main road, the rear vehicle needs to drive at a constant velocity in the acceleration lane and wait for a period of time, so that the front vehicle can complete its longitudinal velocity and position adjustments according to the following model. After the adjustment, the rear vehicle in the sub-platoon will merge into the main road in turn, according to the above mode. During the longitudinal adjustment process, the ultimate goal is that the velocity of the vehicle in front of the sub-platoon is the same as that of the vehicle in front of the gap, and the distance between the vehicle in front of the sub-platoon and the vehicle in front of the gap is the minimum safe distance. The schematic diagram of following–merging rule scenario is shown in Figure 5 as follows.

This rule, in which vehicles in the sub-platoon are merged in turn, has high safety and strong controllability, and the merging process is relatively stable. However, compared with vehicles in the sub-platoon, the total time is longer, and it is suitable for the situation of relatively dense traffic flow on the main road. Following is a scenario diagram of the following–merging rule:

(1)Following model

The intelligent driver model (IDM) proposed by Treiber et al. [31] was used to describe the following behavior of HDVs. The expression is as follows:(7)$$\left\{\begin{array}{c} {\dot{\overset{⌢}{v}}}_{i}\left(t\right)=a_{m}\left[1-\left(\frac{{\overset{⌢}{v}}_{i}\left(t\right)}{v_{f}}\right)-{\left(\frac{S^{*}{}_{\overset{⌢}{i},\overset{¯}{j}}\left(t\right)}{\Delta x_{\overset{⌢}{i},\bar{j}}\left(t\right)}\right)}^{2}\right]\\ S^{*}{}_{\overset{⌢}{i},\bar{j}}\left(t\right)=S_{0}+{\overset{⌢}{v}}_{i}\left(t\right)\cdot T+\frac{{\overset{⌢}{v}}_{i}\left(t\right)\cdot \Delta v_{\overset{⌢}{i},\overset{¯}{j}}\left(t\right)}{2\sqrt{ab}} \end{array}\right..$$
where ${\dot{\overset{⌢}{v}}}_{i}\left(t\right)$ is the acceleration of the following vehicle; $a_{m}$ is the maximum acceleration of the vehicle (which is a constant value related to the dynamic performance of the vehicle); $v_{f}$ is the desired velocity of the following vehicle; $S^{*}{}_{\overset{⌢}{i},\overset{¯}{j}}\left(t\right)$ is the expected headspace between following vehicle and preceding vehicle;$\;\Delta x_{\overset{⌢}{i},\overset{¯}{j}}\left(t\right)$ is the actual headspace between the following vehicle and preceding vehicle; $S_{0}$ is minimum distance between the following vehicle and preceding vehicle; T is the desired time headway; $\Delta v_{\overset{⌢}{i},\overset{¯}{j}}\left(t\right)$ is the velocity difference from the preceding vehicle; b is the desired deceleration.

In addition, the ACC and CACC models were applied to describe the behavior of CAVs following HDVs and CAVs [32,33,34]. The ACC model is written as:(8)$${\dot{\overset{⌢}{v}}}_{i}\left(t\right)=k_{1}\left(\Delta x_{\overset{⌢}{i},\overset{¯}{j}}\left(t\right)-L-S_{0}-t_{a}v\right)+k_{2}\Delta v$$
where $k_{1},k_{2}$ are the control parameters; L is the length of the vehicle (constant value); $t_{a}$ is the expected headway of the following vehicle under the ACC model.

The CACC model expression is:(9)$${\dot{\overset{⌢}{v}}}_{i}\left(t\right)=\frac{k_{p}\left(\Delta x_{\overset{⌢}{i},\overset{¯}{j}}\left(t\right)-L-S_{0}-t_{c}v\right)+k_{d}\Delta v}{k_{d}t_{c}+\Delta t}$$
where $k_{p}{}_{\;},k_{d}$ are the control parameters; $t_{c}$ is the expected headway of the following vehicle under the CACC model; Δt is the time parameter, which is 0.01 s.

(2)Longitudinal adjustment

After longitudinal velocity adjustment by following vehicle, its final velocity is consistent with that of the leading vehicle, and the distance between the heads is the equilibrium distance corresponding to the velocity. In the longitudinal adjustment process of the following vehicle, the following formula is satisfied:(10)$$\left\{\begin{array}{l} {\overset{⌢}{x}}_{i}\left(t+n\cdot \Delta t\right)=\sum_{k=1}^{n}\frac{{\overset{⌢}{v}}_{i}{}^{2}\left[t+n\cdot \Delta t\right]-{\overset{⌢}{v}}_{i}{}^{2}\left[t+\left(n-1\right)\cdot \Delta t\right]}{2{\dot{\overset{⌢}{v}}}_{i}\left(t\right)}-t_{{\overset{⌢}{C}}_{i}}^{la}\cdot v_{main}+S_{\overset{⌢}{i},\bar{j}}^{lat}\\ v_{main}={\overset{⌢}{v}}_{i}(t+n\cdot \Delta t)={\overset{⌢}{v}}_{i}\left(t\right)+\Delta t\;\cdot \sum_{k=0}^{n-1}{\dot{\overset{⌢}{v}}}_{i}\left(t+k\cdot \Delta t\;\right)\\ t_{{\overset{⌢}{C}}_{i}}^{la}=n\cdot \Delta t \end{array}\right.$$
where ${\overset{⌢}{x}}_{i}\left(t+n\cdot \Delta t\right)$ is the lateral position of the vehicle at each moment obtained by RSU; k is the time step count variable, k=1,2,3,…; *n* is the total number of time steps required by the longitudinal adjustment process.

The time required for the vehicle longitudinal adjustment process $t_{{\overset{⌢}{C}}_{i}}^{la}$ can be calculated with the above formula. Considering that the driving distance of the rear car in the sub-platoon is limited by the length of the acceleration lane during the process of waiting to merge, the maximum number of vehicles that can be accommodated in the gap can be expressed as:(11)$$\left\{\begin{array}{l} \Delta x_{ACC}+\left(A_{G_{i}}-1\right)\cdot \Delta x_{CACC}<S_{G_{i}}\\ \left[A_{G_{i}}\cdot t_{lc}+\sum_{i=1}^{q}t_{{\overset{⌢}{C}}_{i+1}}^{la}\right]\cdot v_{ramp}\leq L_{acc}-{\overset{⌢}{x}}_{i}\left(t\right)\\ 0=k_{1}\left(\Delta x_{ACC}-L-S_{0}-t_{a}v_{main}\right)\\ 0=\frac{k_{p}\left(\Delta x_{CACC}-L-S_{0}-t_{c}v_{main}\right)}{k_{d}t_{c}+\Delta t} \end{array}\right.$$
where $\Delta x_{ACC}$ and $\Delta x_{CACC}$ are the balance state (the acceleration of the following vehicle is 0, and the difference in speed is 0) headspace of the ACC and CACC following vehicles, respectively; $S_{G_{i}}$ is the $G_{i}$ gap space distance; $A_{G_{i}}$ is the maximum number of vehicles that can be accommodated in the gap $G_{i}$.

### 2.3. Constraint Condition Statement

In this study, considering the optimal traffic flow safety and efficiency of the on-ramp CAV platoon after the merging of vehicles, and satisfying the conditions for the feasibility of the split strategy and the stability of traffic flow after merging, a performance measurement function was constructed to evaluate all possible split strategies for the platoon, so as to finally obtain the optimal split strategy.

#### 2.3.1. Feasibility Analysis

This section simply lists all of the possible split situations for different sizes of platoons and expresses them mathematically. Combined with the actual space situation for main road clearance, all of the splitting methods conforming to the road conditions were selected.

For a platoon composed of P vehicles, the number set of sub-platoons Q that may be formed after splitting can be expressed as:(12)$$\Omega =\left\{Q|Q=1,2,\cdots ,P;\;Q\in Z\right\}$$

All possible split results should satisfy the following formula:(13)$$\left\{\begin{array}{l} n_{i}<n_{i+1}\\ n_{i+1}=n_{i}+t_{i}\\ t_{i}\in \left[1,P-1\right],t_{i}\in Z\\ n_{0}=0,n_{Q}=p\\ 0\leq i\leq Q-1,1\leq n_{i}\leq P-1 \end{array}\right.$$

In the range of 0≤i≤Q−1 and $1\leq n_{i}\leq P-1$, all feasible solutions of the above inequality correspond to all platoon split results.

In the above equation, $q_{i+1}$ is the scale of the i+1 sub-platoon after splitting. Obviously:$$q_{i+1}=n_{i+1}-n_{i}$$

$q_{1},q_{2},\cdots ,q_{Q}$ can be solved successively for the corresponding $p_{1},p_{2},\cdots ,p_{Q}$ respectively. The vehicle numbers contained in each sub-platoon can be deduced to satisfy the following formula:(14)$$q_{r}=\left\{\begin{array}{l} \left\{{\overset{⌢}{C}}_{i}\;\left|\;{\overset{⌢}{C}}_{1+\sum_{k=1}^{r}p_{{}_{k-1}}}\;,\;\cdots \;,\;{\overset{⌢}{C}}_{\sum_{k=1}^{r}p_{{}_{k}}}\right.\right\}\;,p_{r}\neq 1\\ \left\{{\overset{⌢}{C}}_{i}\;\left|\;{\overset{⌢}{C}}_{\sum_{k=1}^{r}p_{{}_{k}}}\right.\right\}\;,p_{r}=1 \end{array}\right.$$

For the main road clearance set $\left\{G_{1},G_{2},\cdots ,G_{i},\cdots ,G_{I}\right\}$ and a certain platoon split strategy sets $\left\{p_{1}\left\{{\overset{⌢}{C}}_{1},\cdots ,{\overset{⌢}{C}}_{q_{1}}\right\},p_{2}\left\{{\overset{⌢}{C}}_{q_{1}+1},\cdots ,{\overset{⌢}{C}}_{q_{2}}\right\},\cdots ,p_{Q}\left\{{\overset{⌢}{C}}_{q_{Q-1}+1},\cdots ,{\overset{⌢}{C}}_{q_{Q}}\right\}\right\}$ in the information acquisition area, the feasible split strategy for the platoon should satisfy the following constraints:(15)$$\left\{\begin{array}{l} I\geq Q\\ A_{G_{i}{}_{+t_{x}}}\geq q_{x+1} \end{array}\right.$$

See the following Table 1 for parameter descriptions.

Then, a set satisfying the conditions can be obtained, and the mapping of sub-platoons to specific main road clearances can be realized after splitting.

#### 2.3.2. Stability Analysis

In this study, the transfer function method is used to distinguish the stability of the mixed traffic flow formed by merging vehicle platoons. The general form of the vehicle platoon stability transfer function widely accepted in existing research studies is as follows [35,36,37,38]:(16)$$G_{i}\left(s\right)=\frac{U_{i}\left(s\right)}{U_{i-1}\left(s\right)}=\frac{s\cdot f_{i}^{\Delta v}}{s^{2}-s\cdot \left(f_{i}^{v}-f_{i}^{\Delta v}\right)+f_{i}^{h}}$$
where $G_{i}\left(s\right)$ is the disturbance transfer function; s is the Laplacian domain; $U_{i}\left(s\right)$ is the Laplace transform of the velocity perturbation ${\dot{v}}_{i}\left(t\right)$; $f_{i}^{h}$,$f_{i}^{v}$,$f_{i}^{\Delta v}$ is the partial derivative of the corresponding following model with respect to distance, velocity and velocity difference at the equilibrium point, i.e., $f_{i}^{h}=\left.\frac{\partial f}{\partial h}\right|\left(h_{e},v_{e},0\right)$,$f_{i}^{v}=\left.\frac{\partial f}{\partial v}\right|\left(h_{e},v_{e},0\right)$, $f_{i}^{\Delta v}=\left.\frac{\partial f}{\partial \Delta v}\right|\left(h_{e},v_{e},0\right)$, and the expression is shown as follows:

By substituting the partial derivative term in the Table 2 into the disturbance transfer function, the disturbance transfer function expressions for vehicles under the three types of following models, denoted as $G_{IDM}\left(s\right)$, $G_{ACC}\left(s\right)$, and $G_{CACC}\left(s\right)$, respectively, can be obtained as follows:(17)$$G_{IDM}\left(s\right)=\frac{-\sqrt{\frac{a}{b}}v\left[1-{\left(\sqrt{\frac{v}{v_{f}}}\right)}^{4}\right]}{\left(x_{0}+vT\right)\cdot s^{2}+\left\{\frac{4av^{3}\left(x_{0}+vT\right)}{v_{f}{}^{4}}+\left(2aT-\sqrt{\frac{a}{b}}v\right)\cdot \left[1-{\left(\sqrt{\frac{v}{v_{f}}}\right)}^{4}\right]\right\}\cdot s+2a{\left[1-{\left(\sqrt{\frac{v}{v_{f}}}\right)}^{4}\right]}^{\frac{3}{2}}}$$
(18)$$G_{ACC}\left(s\right)=\frac{s\cdot k_{2}}{s^{2}+s\cdot \left(k_{1}t_{a}+k_{2}\right)+k_{1}}$$
(19)$$G_{CACC}\left(s\right)=\frac{s\cdot k_{d}}{\left(k_{d}t_{c}+\Delta t\right)\cdot s^{2}+\left(k_{p}t_{c}+k_{d}\right)\cdot s+k_{p}}$$

Since the propagation of disturbance in the platoon is characterized by the pair propagation of adjacent vehicles, the discriminant conditions for the stability of the platoon can be expressed by the product of the transfer functions of disturbance terms for each vehicle, as follows:(20)$$G\left(s\right)={\left(G_{IDM}\left(s\right)\right)}^{N-1-(P-Q)}{\left(G_{ACC}\left(s\right)\right)}^{Q}{\left(G_{CACC}\left(s\right)\right)}^{P-Q}$$
where $G\left(s\right)$ is the overall disturbance transfer function of mixed traffic flow formed after intelligent platoon merging.

Let s=jω, then the transfer function is transformed from the Laplacian domain to the frequency domain, and according to the transfer function theory, the stability discrimination condition of mixed traffic flow is obtained as follows:(21)$${\left\|G\left(s\right)\right\|}_{\infty }={\left\|{\left(G_{IDM}\left(s\right)\right)}^{N-1-(P-Q)}{\left(G_{ACC}\left(s\right)\right)}^{Q}{\left(G_{CACC}\left(s\right)\right)}^{P-Q}\right\|}_{\infty }\leq 1,\forall \omega >0$$
where $\left|\;\cdot \;\right|$ is the amplitude in the frequency domain of the transfer function, and j and ω are an imaginary number and frequency in the frequency domain, respectively.

#### 2.3.3. Safety Analysis

In this study, the Time-Exposed Rear-end Crash Risk Index (TERCRI) was used to describe the degree of risk of a rear-end collision caused by an insufficient safety distance between the front and rear vehicles due to the current rapid deceleration of vehicle velocity [39,40]. The calculation formula is as follows:(22)$$TERCRI\left(t\right)=\sum_{i=1}^{N}\sum_{t=0}^{M}RCRI_{i}\left(t\right)\cdot \Delta t=J_{sa}$$
(23)$$RCRI_{n}\left(t\right)=\left\{\begin{array}{l} 1,D_{F}>D_{L}\\ 0,D_{F}\leq D_{L} \end{array}\right.$$
where $J_{sa}$ is the safety index, and $D_{L}$ and $D_{F}$ are the stopping distance of the front and rear vehicles, respectively, which can be calculated as:(24)$$\left\{\begin{array}{l} D_{L}=v_{i-1}h+\frac{v_{i-1}^{2}}{2b}+L\\ D_{F}=v_{i}t_{PRT}+\frac{v_{i}^{2}}{2b} \end{array}\right.$$
where $t_{PRT}$ is the perceived response time.

#### 2.3.4. Efficiency Analysis

An efficiency index that considers the average travel time of all vehicles in the CAV platoon is defined as $J_{e}$. $J_{e}$ is shown as:(25)$$J_{e}=\frac{1}{P}\left[\sum_{i=1}^{P}t_{i}^{fin}-t_{i}\left(x_{O}\right)\right]$$
where $t_{i}^{fin}$ is the completion time of vehicle merging into the mainline traffic, $t_{i}\left(x_{O}\right)$ is the moment when the vehicle ${\overset{⌢}{C}}_{i}$ reaches the start of the acceleration lane.

#### 2.3.5. Performance Measurement

Considering feasibility, traffic flow stability, safety and vehicle merging efficiency comprehensively, the evaluation function is constructed as follows:(26)$$J=\xi_{1}J_{e}+\xi_{2}J_{sa}$$
(27)$$s.t.\left\{\begin{array}{l} {\left\|G\left(s\right)\right\|}_{\infty }\leq 1\\ L+S_{0}+t_{a}{\overset{⌢}{v}}_{i}+\left(A_{G_{i}}-1\right)\cdot \left(L+S_{0}+t_{c}{\overset{⌢}{v}}_{i}\right)<S_{G_{i}}\\ {}[(A_{G_{i}}-1)\cdot t_{lc}+\sum t_{{\overset{⌢}{C}}_{i+1}}^{la}]\cdot v_{ramp}\leq L_{acc}-{\overset{⌢}{x}}_{i}\left(t\right)\\ A_{G_{i}{}_{+t_{x}}}\geq q_{x+1}\\ 0\leq x\leq Q-1,x\in Z\\ 1\leq t_{x}\leq I-1,t_{x}\in Z\\ t_{x+1}>t_{x}\\ I\geq Q\\ \sum q_{i}=P\\ \xi_{1}+\xi_{2}=1 \end{array}\right.$$
where $\xi_{1}$ and $\xi_{2}$ are the weight values of the efficiency index and safety index, respectively.

The optimal solution of the above equation can be selected to obtain the optimal platoon split strategy under the current traffic condition. The platoon split strategy includes the number of sub-platoons formed after the split and the vehicle numbers contained in each sub-platoon, which can be expressed as follows:(28)$$\left\{p_{1}\left\{{\overset{⌢}{C}}_{1},\cdots ,{\overset{⌢}{C}}_{q_{1}}\right\},p_{2}\left\{{\overset{⌢}{C}}_{q_{1}+1},\cdots ,{\overset{⌢}{C}}_{q_{2}}\right\},\cdots ,p_{Q}\left\{{\overset{⌢}{C}}_{q_{Q-1}+1},\cdots ,{\overset{⌢}{C}}_{q_{Q}}\right\}\right\}$$

## 3. Experiment Design

To validate the effectiveness of the proposed strategy, a case study was designed. The case study utilized traffic simulation data. Firstly, a simulation environment was built based on the Python programming language, as shown in Figure 6. In this environment, a merging area consisting of a one-lane ramp and a three-lane mainline freeway was set. The X-axis was set as the center line of the middle lane, with the right direction as the traffic flow direction and the positive direction. Additionally, the alternate area of the main insertable gap extends from 675 m to 975 m, with a total length of 300 m. Moreover, ramp vehicles can use the gaps in the main lane within the alternate area to complete merging.

In this simulation environment, the traffic flow on the main road is set to 1500 pcu/h/lane, and vehicles arrive randomly according to the Poisson distribution. The selection of 1500 pcu/h/lane is because it is slightly higher than the designed flow rate of 1200 pcu/h/lane on a freeway, which means that the traffic volume is heavy on the main road. In this state, the risk of on-ramp vehicle merging under the control of a traditional strategy increases, and the efficiency decreases, which highlights the improved efficiency and safety of the strategy proposed in this study. The simulation vehicle is 4.3 m long and 1.8 m wide (a typical passenger vehicle size). The total simulation time is 150 s, and the data are collected every 0.1 s. The particular parameters of the IDM model, ACC model and CACC model are listed as follows.

(i).IDM: The expected acceleration and deceleration of the HDVs are 5 $m/s^{2}$ and −5 $m/s^{2}$, respectively, with a time headway of 1.8 s and minimum parking distance of 5 m.(ii).ACC model: The related parameters are set as $k_{1}=0.23$, $k_{2}=0.07$ and $t_{a}=1.1\;s$.(iii).CACC model [35]: The related parameters are set as, $k_{p}=0.45$, $k_{d}=0.25$ and $t_{c}=0.6\;s$.

It is worth mentioning that the CACC model was used to describe the following behavior of the following CAV relative to the front CAV in the simulation experiment, including the generated CAV in the mixed traffic flow on the main road and the continuous CAV formed after the ramp CAV merged into the main road gap.

In addition, some parameter values are independent of the following model. For vehicle dynamic performance, the comfortable acceleration is 3 $m/s^{2}$ while the max acceleration and deceleration are 5 $m/s^{2}$ and −8 $m/s^{2}$, respectively, for all vehicles. For the vehicle driving behavior parameter, the lane changing time is set at 3 s. Considering that the proposed PSCM method is both safe and efficient, the coefficient of driving efficiency index $\xi_{1}$ is set to greater than the safety index $\xi_{2}$. The weighting factors of the evaluation function are set as $\xi_{1}=0.6$ and $\xi_{2}=0.4$.

In order to verify the effectiveness of the proposed strategy, two kinds of simulation scenarios were designed. One is the single-vehicle merging scenario, and the other is the platoon split merging scenario proposed in this study.

(i).In the single-vehicle merging scenario, a platoon composed of four CAVs approaches the starting point of the acceleration lane on the on-ramp and is going to merge into the mainline traffic flow. During this period, no collaborative control at the platoon level is carried out. Instead, it is regarded as four independent CAVs. RSUs analyze collision avoidance conditions and determine the acceptable gap according to the state of the mainline vehicles. Finally, a suitable driving strategy is formulated to complete the merging process. During this process, the vehicle behind the platoon that has not completed merging will regard the vehicle in front as an obstacle vehicle.(ii).In the platoon split merging scenario proposed in this study, a platoon composed of four CAVs can cooperate with the RSUs before entering the acceleration lane to complete the platoon split in advance according to the gap state of the mainline and the driving state of the platoon itself. It is worth noting that the meaning of platoon split at this time focuses on the differences in vehicle control logic in the platoon, rather than the separation of spatial distance between vehicles. After splitting, several small-scale sub-platoons are formed. Each sub-platoon corresponds to an acceptable mainline gap and a set of merging driving control schemes. Each vehicle in the sub-platoon executes in turn to complete the merging process.

The entire simulation experiment process included the following three steps:

Step 1. The single-vehicle merging scenario experiment. In this scenario, the initial ratio of CAVs to HDVs in the main road traffic flow was set at 50%, the mainline traffic input was set at 1500 pcu/h/lane, and the vehicles arrived randomly according to the Poisson distribution. The single simulation test lasted for 3 min, and the simulation step length was 0.1 s. After the traffic flow on the main road stabilizes, a platoon of four CAVs is generated at the start of the on-ramp to move at a constant velocity of $\overset{⌢}{v}=60\;\mathrm{km}/h$ with $t_{c}=0.6\;s$ into the acceleration lane. If the safety conditions for collision avoidance are met, the merging operation is carried out. The vehicle number, velocity, acceleration and position coordinate information of all main road and on-ramp vehicles passing through the alternative area of the main road gap in each time step are recorded.

Step 2. The platoon split merging scenario experiment. In this scenario, the initial settings, such as mixed traffic flow ratio, main line flow input and ramp platoon generation, are the same as those of the single-vehicle merging scenario. Considering all possible split strategies (except 1-1-1-1), there are seven possible split strategies for four vehicles, namely 0-4, 1-3, 2-2, 3-1, 1-1-2, 1-2-1, 2-1-1. Each strategy is executed according to the merging vehicle motion control rules. The vehicle number, velocity, acceleration and position coordinate information of all main road vehicles and ramp platoons passing through the alternative area of the main road gap in each time step are recorded. To simplify the expression, each split strategy is represented by a corresponding letter, as shown in the following Table 3:

Step 3. The traffic flow environment is initialized with different datasets consisting of 50% and 70% ratios of CAVs to HDVs. The aforementioned Step 1 and Step 2 are repeated. In the same way, the velocity, acceleration and position data of all vehicles during the entire time in the diverging area are recorded as more persuasive information.

Remarks: (i) Split strategy (x-y) indicates that according to the driving direction, first, (x) vehicles form a sub-platoon, and next, (y) vehicles form another sub-platoon.

(ii) The same split strategy is represented by $\overset{⌢}{a}$ and $\bar{a}$ with the CAV ratios of 30% and 70%, respectively.

## 4. Results Analysis

According to the experimental design in Section 3, the proposed platoon split strategy was simulated. Based on the simulation data obtained from the experiment, the spatiotemporal trajectory diagram and velocity and acceleration change diagram of the CAV platoon and related vehicles before and after the main road clearance under the control of the platoon split strategy and without the control of the platoon splitting strategy were drawn. Take the ratio of 50% as an example, as shown in the figure below. In the figure, the gray area in the spatiotemporal trajectory diagram represents the spatial position of the acceleration lane in the merging area. In addition, the figure numbered (×1) is the spatiotemporal trajectory diagram of four CAVs and the vehicles before and after the main road gap under the control of a certain strategy. Similarly, the figures numbered (×2) and (×2) are the velocity and acceleration changes of four CAVs under the control of this strategy, respectively.

According to the definition in Table 3, plot the spatiotemporal trajectory, velocity and acceleration under 50% CAV ratio corresponding to the eight split strategies as shown in Figure 7, that is, Figure 7(a1–a3) are spatiotemporal trajectory, velocity and acceleration figures respectively under the no-split strategy, Figure 7(b1–b3) are spatiotemporal trajectory, velocity and acceleration figures respectively under the 0-4 split strategy, Figure 7(c1–c3) are spatiotemporal trajectory, velocity and acceleration figures respectively under the 1-3 split strategy, Figure 7(d1–d3) are spatiotemporal trajectory, velocity and acceleration figures respectively under the 2-2 split strategy, Figure 7(e1–e3) are spatiotemporal trajectory, velocity and acceleration figures respectively under the 3-1 split strategy, Figure 7(f1–f3) are spatiotemporal trajectory, velocity and acceleration figures respectively under the 1-1-2 split strategy, Figure 7(g1–g3) are spatiotemporal trajectory, velocity and acceleration figures respectively under the 1-2-1 split strategy, and Figure 7(h1–h3) are spatiotemporal trajectory, velocity and acceleration figures respectively under the 2-1-1 split strategy.

Figure 7(a1–a3) show the spatiotemporal trajectory of the merging CAV and the front and rear vehicles on the main road, as well as the changes in velocity and acceleration of the merging CAV under no-split control strategy. As can be seen from the spatiotemporal trajectory diagram of Figure 7(a1), the merging vehicles ONRAMP 8 and ONRAMP 11 both have parking phenomena, and the parking location is close to the downstream of the acceleration lane, which means the two CAVs spent a long time in the acceleration lane and failed to merge. In addition, the acceleration behavior of ONRAMP 8 has a significant impact on the rear main road vehicles, forcing them to significantly slow down. This means that under no-split control strategy, some merging vehicles have problems such as low efficiency, great negative impact on the main road traffic and high accident potential. Therefore, different platoon split strategies can be considered to improve the effect of CAV platoon merging.

It is shown in the Figure 7 above that the vehicle trajectory under different platoon split strategies is quite different and shows different merging effects. Specifically analyzing the merging performance of a CAV platoon under the control of different strategies, it can be found that some vehicles still have stopping behavior in some strategies, such as 0-4, 1-3, and 3-1, which is represented by the horizontal part of the spatiotemporal diagram (b1, c1 and e1) and the zero point of the velocity change diagram (b2, c2, and e2). Taking the split strategy 1-3 as an example, both ONRAMP 9 and ONRAMP 10 have stopping time, and the maximum parking time reaches 12 s (seen from c1). The reason is that when there are too many vehicles in one sub-platoon, they need to stop and wait for a considerable gap in the main road traffic flow for merging. Comparatively, the merging under the other split strategies (such as 2-2 and 1-1-2) has better performance, which is reflected in the following aspects:(i).The range of change in velocity is small (seen from d2, f2) and the frequency of change in acceleration is low (seen from d3, f3), which represents a smooth driving process, high safety and high driving comfort.(ii).No stopping behavior is observed. Moreover, the merging time is short, and the vehicles in front and behind on the main road do not slow down significantly (seen from d1, f1).

In addition, it can be seen from Figure 7(g2,g3,h2,h3) that although no stopping occurs for the merging vehicles under the control of the split strategies 1-2-1 and 2-1-1, the velocity curves show that the vehicles have many acceleration and deceleration fluctuations, and the acceleration curves also have many peaks and troughs, such as for ONRAMP8 in (g2), where three decelerations occurred in 10 s before the acceleration for merging, but the deceleration amplitude was small, no more than 5 m/s; ONRAMP9 in (h2) had two decelerations before the acceleration for merging, but the deceleration amplitude was large, and the maximum was more than 15 m/s. This means that the driving stability during the merging process was poor, with frequent acceleration and deceleration also leading to decreased comfort.

In summary, two conclusions can be drawn from the above analysis:(i).The platoon split merging strategy proposed in this study can improve the efficiency and safety of platoon merging in the merging area under certain circumstances and reduce the impact on the main road vehicles.(ii).The final merging performance of the CAV platoon is different under the control of different split strategies.

Simulation experiments with the main route CAV ratios of 30%, 50% and 70% were respectively set in order to analyze the influence of the CAV ratio on the split strategy. The 30%, 50% and 70% CAV ratios were selected to correspond to the three stages of CAV popularization: basic ratio, relatively high ratio and basic large-scale popularization. Taking the 2-2 split strategy as an example, the merging performance in three simulation experiments is shown in Figure 8 below:

In Figure 8, ($\overset{⌢}{d}1$), ($\overset{⌢}{d}2$) and ($\overset{⌢}{d}3$) in the first column on the left respectively indicate the spatiotemporal trajectory, velocity and acceleration figures in the 2-2 split strategy under 30% CAV ratio. Similarly, (d1), (d2) and (d3) in the second column indicate the spatiotemporal trajectory, velocity and acceleration figures in the 2-2 split strategy under 50% CAV ratio. ($\bar{d}1$), ($\bar{d}2$) and ($\bar{d}3$) in the third column indicate the spatiotemporal trajectory, velocity and acceleration figures in the 2-2 split strategy under 70% CAV ratio. It can be roughly seen from the spatiotemporal trajectory diagram that the overall effect of the 2-2 split strategy was relatively good under the three CAV ratios.

(i).For the merging vehicles, the velocity variation chart has no zero point, indicating that the vehicles have no stopping behavior. The four spatiotemporal trajectory curves are generally smooth and compact, indicating that the vehicles run smoothly, the distance between the heads is small, and the merging efficiency is high.(ii).For vehicles on the main road, the spatiotemporal trajectory curves are generally straight, and the slope changes are small, indicating that the main road vehicles have no significant deceleration behavior and are less affected by the platoon merging behavior.

However, differences can still be found by analyzing the velocity–acceleration curve of the vehicles under the three CAV ratios. Firstly, in general, with the increase in CAV ratios, the performance of platoon entry was significantly improved. In terms of velocity change, taking the last merging vehicle as an example, under the 30% CAV ratio, the range of velocity change was large, up to 30 m/s. In terms of acceleration change, the acceleration time of each vehicle was long, and the maximum acceleration time was 6 s. In contrast, under the 50% and 70% CAV ratios, the maximum range of velocity change was 20.4 m/s and 5.5 m/s, respectively, and the maximum acceleration time was 4.9 s and 5.1 s, respectively, which were significantly improved. In addition, it can also be seen intuitively from the acceleration change curve that the time taken by all vehicles in the platoon from chasing the gap to reaching stable vehicle velocity after the completion of merging decreases with the increase in CAV ratio, indicating to a certain extent that the merging efficiency increases with the increase in the CAV ratio.

By comparing the strategy charts under the 50% ratio, it can be seen that there is a split strategy with the optimal merging effect, which makes the platoon merge efficiency high and velocity changes gentle and has little influence on the normal flow of the main road vehicles. Furthermore, in order to explore the influence of the CAV ratio on the optimal split strategy, the velocity and acceleration curves of merging vehicles under the corresponding optimal split strategy were drawn under the three ratio conditions of 30%, 50% and 70% as shown in Figure 9. The results showed that at 30%, 50% and 70% ratios, the optimal split strategies were 1-3, 2-2, and 2-2, respectively.

In Figure 9, ($\overset{⌢}{c}2$) and ($\overset{⌢}{c}3$) in the first column on the left respectively indicate the velocity and acceleration figures in the 1-3 split strategy under 30% CAV ratio. Similarly, (d2) and (d3) in the second column indicate the velocity and acceleration figures in the 2-2 split strategy under 50% CAV ratio. ($\bar{d}2$) and ($\bar{d}3$) in the third column indicate the velocity and acceleration figures in the 2-2 split strategy under 70% CAV ratio. The above optimal splitting strategies have some common points:(i).The number of vehicles in the sub-platoon should not be too large. It is mainly reflected in the increase in stopping time caused by waiting for a large gap, and in the process of stopping and starting, the car behind on the main road is forced to slow down significantly.(ii).The number of sub-platoons should not be too many. It is mainly reflected in the velocity disturbance, which amplifies the transmission between platoons, resulting in a decrease in the driving stability of downstream vehicles. In summary, there is a contradiction between the two points. Therefore, under specific ratios, the optimal split strategy for the arrival of the platoon is different, which is related to the gap distribution and the parameters of the platoon and needs to be analyzed in detail.

From the perspective of merging travel time, the application of the optimal split strategy significantly improves the merging efficiency of the platoon as shown in Table 4. Taking the 50% ratio as an example, the merging travel time of each vehicle in the platoon under the no-control strategy and seven platoon split strategies was calculated, as shown in the table. As can be seen from the table, under the 2-2 split strategy, the merging time of vehicles is the shortest, at 10.6 s, and 50.7% lower than that under the no-split control strategy. Under the 1-1-2 and 2-1-1 split control strategies, the average travel time of vehicles is also smaller than that without a control strategy, at 11.75 s and 14.4 s, respectively. On the other hand, it can be found that when the platoon is not split to merge as a whole, due to the stopping and waiting process, the average time duration of vehicle merging is 34.675 s, which increases by 61.3% compared with the average duration of vehicles without strategic control and is much higher than the merging time under other strategies.

The evaluation of the platoon split strategy should also take into account the impact of merging vehicles on adjacent vehicles on the main road under the strategy. Therefore, the average velocity, standard deviation of velocity, average acceleration, standard deviation of acceleration and average travel time of vehicles before and after the main road clearance within the range of influence (950 m–1150 m) under the three CAV ratios were calculated to explore the influence of different strategies on the flow of vehicles on the main road. The summary Table 5 is as follows:

As can be seen from Table 5, when the CAV ratio is 30%, the average velocity of vehicles near the main road without a control strategy is 18.97 m/s, which is only lower than the average velocity under the 0-4 and 3-1 split strategies. Further, the average standard deviations of velocity and acceleration are lower than the mean value of each strategy, which means that without a control strategy, the impact of the vehicle merging on the main road is small, and the effect is relatively good. However, in summary, under the 1-3 split strategy, although the average velocity is only 0.56% lower than that without strategic control, the average standard deviation of velocity is 15.82% lower, and average acceleration standard deviation is 26.15% lower than that without strategic control, which means that vehicles on the main road slow down less, and the driving process is more stable and smoother. Therefore, under the 1-3 split strategy, the merging vehicles have the least interference with vehicles on the main road.

Combined with the relevant data for the 30% CAV ratio, it can be concluded that the application of the PSCM method is more helpful in reducing the impact of platoon merging on the main road vehicles as the ratio increases. From Table 4, it can be calculated that at the 70% CAV ratio, the average velocity under the 2-2 split strategy is 23.97% higher than that without strategic control, and the average velocity and acceleration standard deviation are 76.92% and 70.56% lower, whereas the same indexes under the 50% CAV ratio are 22.57%, 72.57% and 68.57%, respectively. In summary, under the 70% CAV ratio, the average velocity of main road vehicles in the 2-2 split strategy increased by 1.40%, and the average standard deviation of velocity and the average standard deviation of acceleration decreased by 4.35% and 1.99%, respectively, compared with 50% CAV ratio. It is indicated that the increase in the CAV ratio can be helpful to further improve the effect of the proposed PSCM method.

## 5. Conclusions and Future Work

This paper focused on the problems of congestion and potential risk as multiple vehicles merge into the merging area of a freeway. We proposed a platoon split collaborative merging method (PSCM) for an on-ramp CAV platoon in the merging area considered safety and efficiency. Firstly, a merging vehicle motion control strategy based on the following–merging rule was established to describe the merging process for on-ramp vehicles. Then, a merging effect evaluation model was constructed considering various factors comprehensively, and the optimal safety and efficiency platoon split strategy was selected. Finally, numerous simulation experiments with different CAV ratios were conducted to verify the effectiveness of the proposed method.

Based on the above traffic simulation experiments and discussion, the following conclusions can be established:(i).For the merging CAV platoon, the PSCM method not only ensures safety during the acceleration and lane changing process, but also significantly promotes the efficiency of the merging CAV platoon, resulting in a 50.7% reduction in the travel time compared with that under the no-split control strategy.(ii).For the main road vehicles, the PSCM method reduces the interference of the CAV platoon with the normal flow of the main road vehicles during the merging process. The experimental results show that under the 50% CAV ratio, the average travel time of main road vehicles under the optimal platoon split strategy is reduced by 27.9%, and the standard deviation of acceleration is reduced by 68.6%.(iii).The PSCM method is more effective in reducing the merging interference with a higher ratio of CAVs in the mixed traffic flow. As the CAV ratio increases from 50% to 70%, the average velocity of main road vehicles in the 2-2 split strategy increases by 1.40%, and the average standard deviation of velocity and the average standard deviation of acceleration decrease by 4.35% and 1.99%, respectively.

However, this study still has certain limitations. In this study, the CAVs in the mixed traffic flow on the main road normally follow, without considering the cooperative control between the CAVs and the on-ramp merging CAV platoon. In future research, the CAV platoon split strategy can be combined with the active control of CAV velocity on the main road to further improve the control effect.

## Figures and Tables

**Figure 1 sensors-23-04401-f001:**
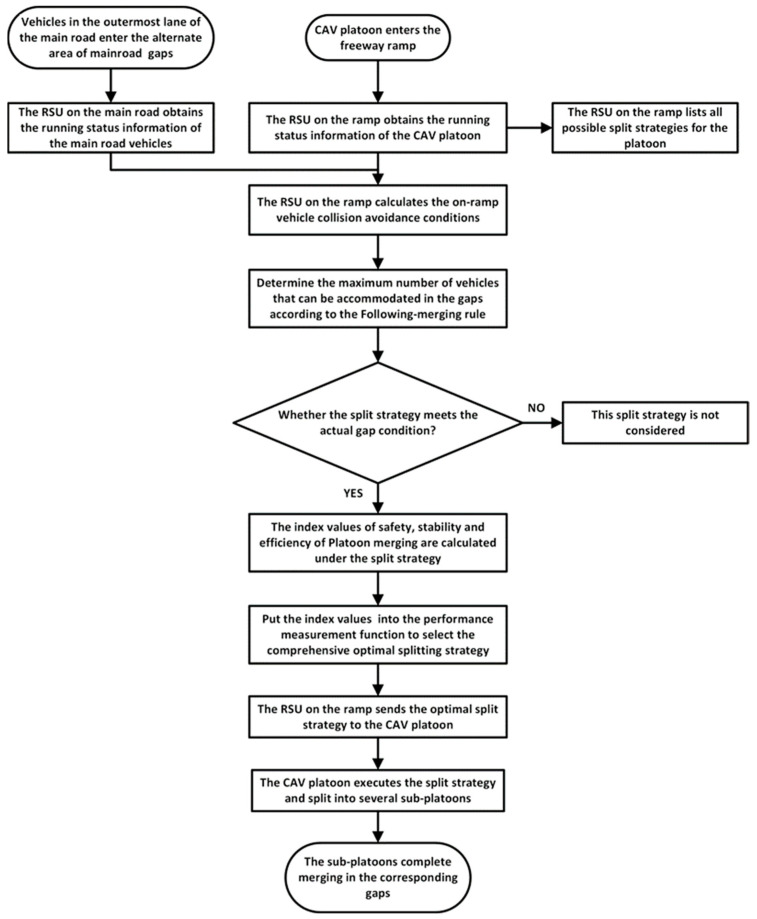
Flowchart of the proposed platoon split collaborative merging (PSCM) method.

**Figure 2 sensors-23-04401-f002:**
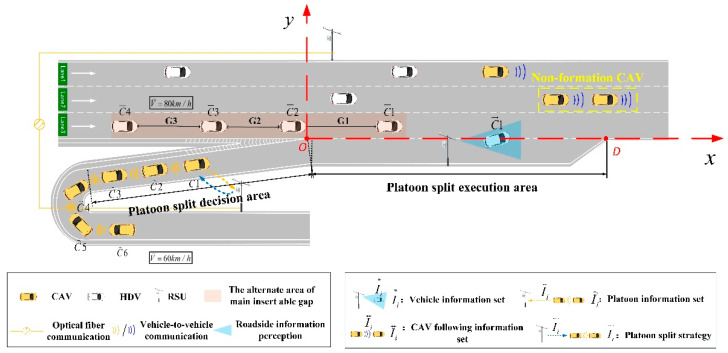
Schematic of the merging area.

**Figure 3 sensors-23-04401-f003:**
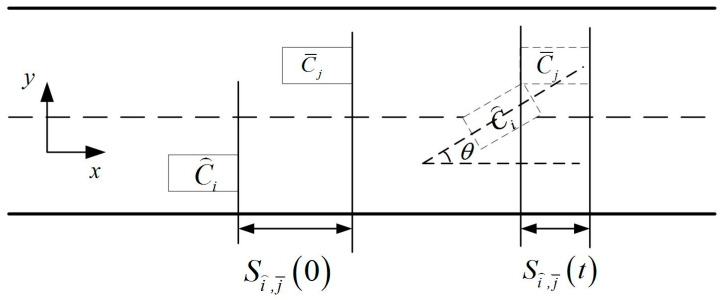
Collision avoidance diagram of the ramp vehicle and the gap ahead vehicle.

**Figure 4 sensors-23-04401-f004:**
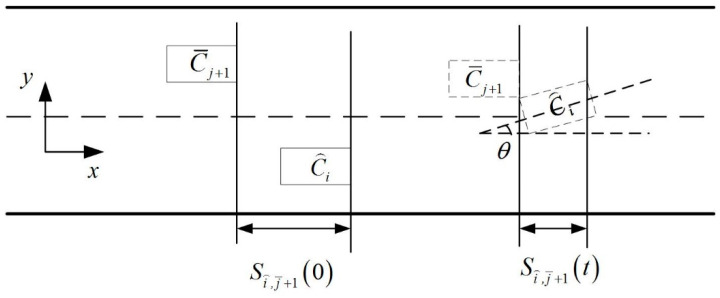
Collision avoidance diagram of the ramp vehicle and the gap behind vehicle.

**Figure 5 sensors-23-04401-f005:**
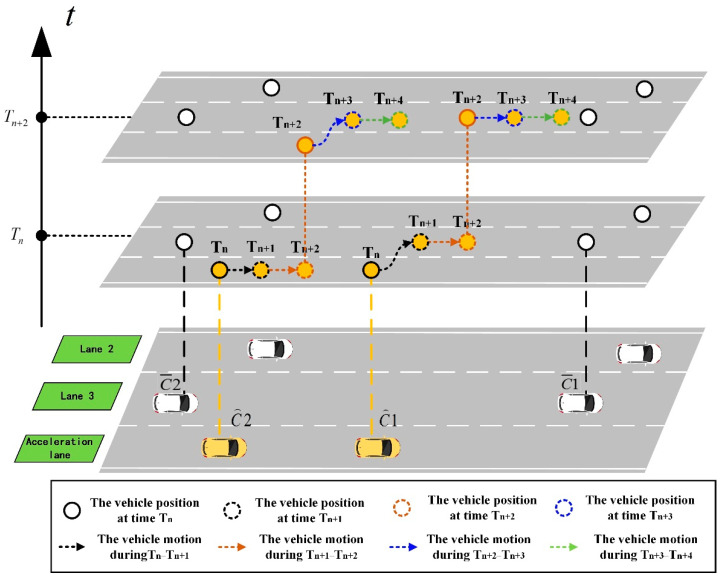
Schematic diagram of following–merging rule scenario.

**Figure 6 sensors-23-04401-f006:**
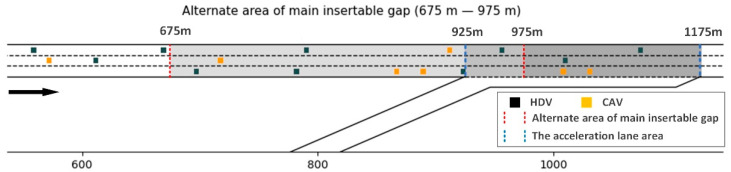
Traffic simulation environment.

**Figure 7 sensors-23-04401-f007:**
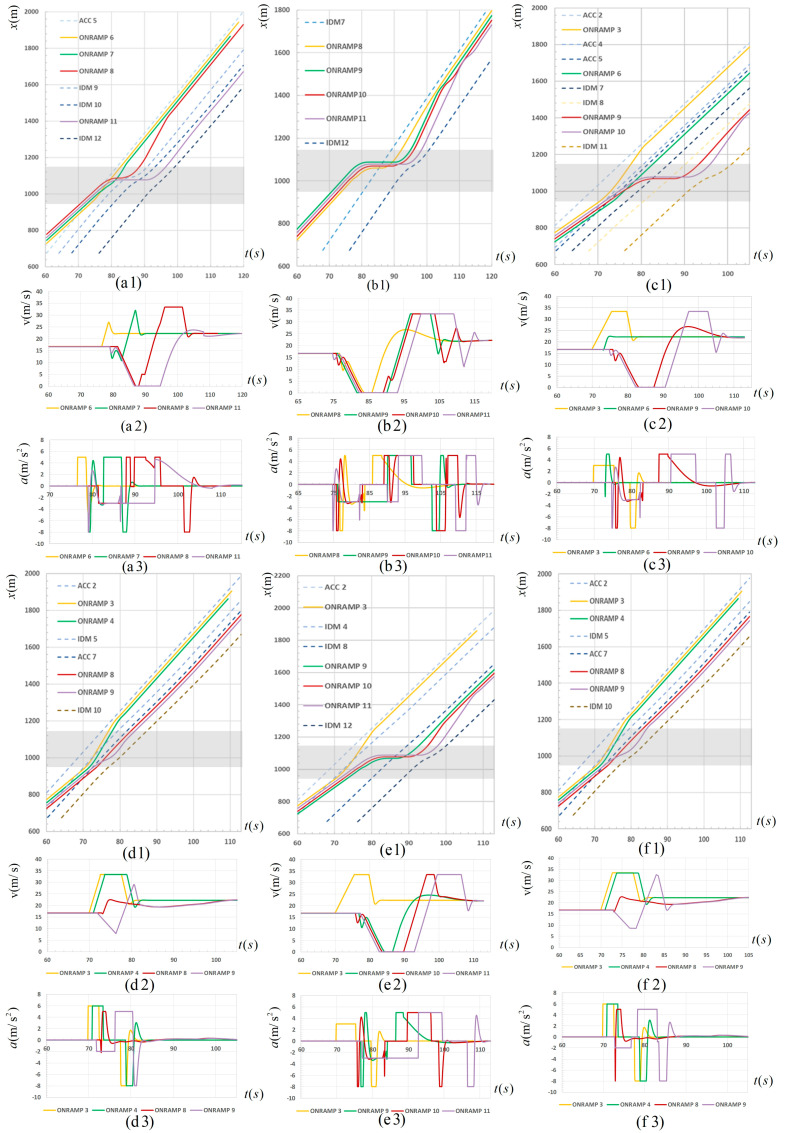
Spatiotemporal trajectory, velocity and acceleration under 50% CAV ratio.

**Figure 8 sensors-23-04401-f008:**
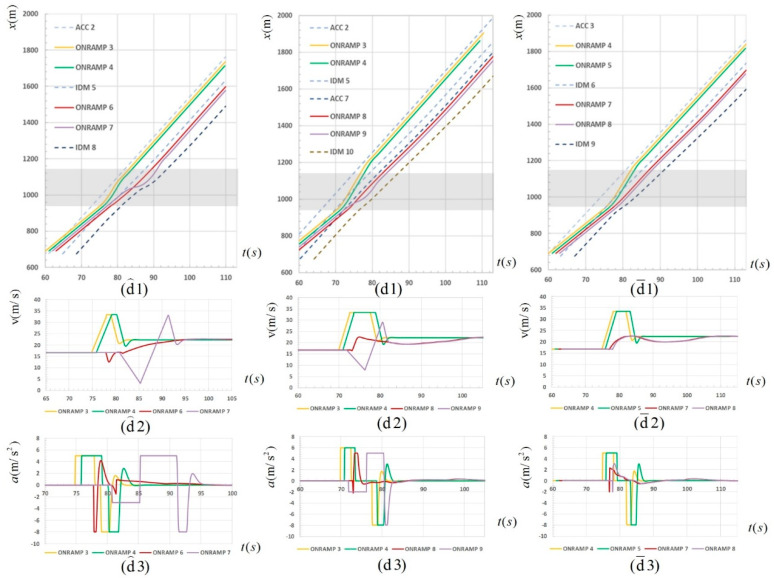
The spatiotemporal trajectory, velocity and acceleration in the 2-2 split strategy under 30%, 50% and 70% CAV ratios.

**Figure 9 sensors-23-04401-f009:**
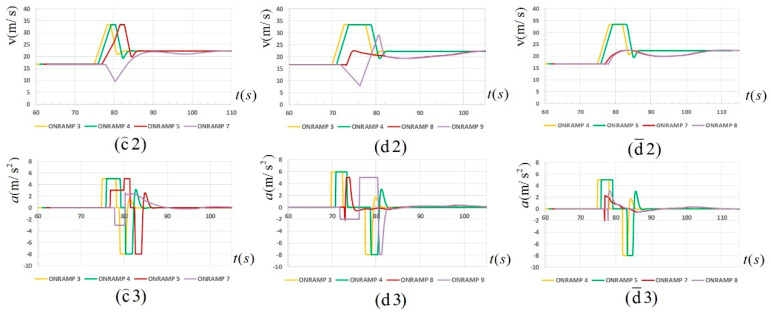
The velocity and acceleration of the optimal split strategy under 30%, 50% and 70% CAV ratios.

**Table 1 sensors-23-04401-t001:** Description of constraint parameters of feasible split strategy for platoons.

Parameter	Remarks	Parameter	Remarks
I	Number of main road gaps	Q	Number of sub-platoons after splitting
i	$i\in \left[1,I\right],\;i\in Z$	$t_{x}$	$t_{x}\in \left[1,I-1\right],\;t_{x}\in Z,\;t_{x+1}>t_{x}$
x	$x\in \left[0,Q-1\right],x\in Z$	$A_{G_{i}}$	The maximum number of vehicles that can be accommodated in the main road gap i

**Table 2 sensors-23-04401-t002:** Partial derivative value expression description.

Model	IDM	ACC	CACC
$f_{i}^{h}$	$2a\cdot \frac{{\left[1-{\left(\frac{v}{v_{f}}\right)}^{4}\right]}^{\frac{3}{2}}}{x_{0}+vT}$	$k_{1}$	$\frac{k_{p}}{k_{d}t_{c}+\Delta t}$
$f_{i}^{v}$	$-\frac{4av^{3}}{v_{f}{}^{4}}-2aT\cdot \frac{1-{\left(\frac{v}{v_{f}}\right)}^{4}}{x_{0}+vT}$	$-k_{1}t_{a}$	$\frac{-k_{p}t_{c}}{k_{d}t_{c}+\Delta t}$
$f_{i}^{\Delta v}$	$-\sqrt{\frac{a}{b}}\cdot \frac{v\cdot \left[1-{\left(\frac{v}{v_{f}}\right)}^{4}\right]}{x_{0}+vT}$	$k_{2}$	$\frac{k_{d}}{k_{d}t_{c}+\Delta t}$

**Table 3 sensors-23-04401-t003:** Letters corresponding to split strategies (50% CAV ratio).

Split Strategy	None	0-4	1-3	2-2	3-1	1-1-2	1-2-1	2-1-1
Letter	a	b	c	d	e	f	g	h

**Table 4 sensors-23-04401-t004:** The merging travel time of CAV platoon under 50% CAV ratio.

Strategies	None	0-4	1-3	2-2	3-1	1-1-2	1-2-1	2-1-1
CAV1 (s)	6.10	19.70	11.90	9.80	11.90	9.80	11.90	11.90
CAV2 (s)	16.10	35.60	12.70	10.40	19.90	10.40	14.40	14.40
CAV3 (s)	33.50	38.70	32.20	12.70	27.40	12.70	9.70	14.40
CAV4 (s)	30.30	44.70	36.20	9.50	38.30	14.10	19.30	16.90
Average Travel Time (s)	21.50	34.675	23.25	10.60	24.375	11.75	13.825	14.40

**Table 5 sensors-23-04401-t005:** The influence of different strategies on the main road vehicles.

Ratio	Index	None	0-4	1-3	2-2	3-1	1-1-2	1-2-1	2-1-1	Average
30%	Average velocity	18.97	25.36	18.48	18.20	24.01	17.99	18.11	18.13	19.91
	Velocity SD	1.96	2.13	1.65	2.39	1.87	2.98	2.93	2.54	2.31
	Average acceleration	0.12	0.52	0.46	0.29	0.65	0.44	0.51	0.30	0.41
	Acceleration SD	1.30	1.71	0.96	1.64	0.96	1.70	1.66	1.71	1.46
	Average travel time	10.78	8.01	10.84	11.09	8.58	11.25	11.15	11.14	10.36
50%	Average velocity	16.57	19.42	20.58	20.34	20.09	20.14	20.32	20.36	19.73
	Average velocity SD	3.61	2.26	1.14	0.99	1.78	1.36	1.34	0.98	1.68
	Average acceleration	−0.08	−0.03	0.05	−0.11	−0.10	−0.16	−0.07	−0.01	−0.06
	Average acceleration SD	2.10	1.25	0.59	0.66	1.17	0.93	1.03	0.76	1.06
	Average travel time	13.05	10.52	9.84	9.86	10.04	9.97	9.94	9.91	10.39
70%	Average velocity	16.56	19.83	18.45	20.53	20.37	20.56	20.37	20.57	19.66
	Average velocity SD	3.25	1.83	2.76	0.75	0.82	0.77	0.83	0.75	1.47
	Average acceleration	−0.06	−0.03	0.24	0.04	0.11	0.04	0.11	0.04	0.06
	Average acceleration SD	1.80	1.03	1.32	0.53	0.43	0.53	0.43	0.55	0.83
	Average travel time	12.09	10.23	11.08	9.79	9.91	9.78	9.90	9.77	10.32

SD refers to standard deviation.

## Data Availability

The data used to support the findings of this study are included within the article.

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
