# Peer review of "A Collaborative Merging Method for Connected and Automated Vehicle Platoons in a Freeway Merging Area with Considerations for Safety and Efficiency"

_sensors, 2023, doi:10.3390/s23094401_

Round 1
Reviewer 1 Report
This article proposes a platoon split collaborative merging method (PSCM) based on mixed traffic environment to address the problems of congestion and accidents in highway merging areas. Using this method, experiments were conducted on mixed traffic with intelligent connected vehicle (CAV) ratios of 30%, 50%, and 70%. The experimental results showed that compared to not using any merging strategy, using the PSCM method for merging can significantly improve the driving efficiency of main road vehicles and merging vehicles. Simultaneously calculate the minimum safe distance between the merging vehicles and the front and rear vehicles in main road to ensure the safety of merging.
The research method of the article is specific and feasible, and has significant research significance. However, there are the following issues, and it is recommended to make moderate modifications:
1. There are spelling errors in the article, such as' mainrod 'and' performance s' located on lines 176, 178, and 398, respectively, in sentences 'Lower boundary of the mainrod gap:', 'Up boundary of the mainrod gap:', and 'For vehicle dynamic performance s'.
2. The article has obvious confusion in sentence structure, such as 'the he positive direction' in line 380.
3. The article mentions in line 195 that 'there is no deceleration behavior during the merging process' is clearly inconsistent with the experimental results in the following text. And what are the parameters for the default safety distance mentioned in line 196, 'the spacing between the front and back of the vehicles in the formation is the safety spacing by default'?
4. What do the meanings of S0, Tcri, Tfin in formulas 4 and 5 represent?
5. The definition of safety indicator Jsa in formula 26 is not reflected in the article.
6. According to the definition of formula 26, ξ1 and ξ2 and should be the weight parameters of efficiency indicators and safety indicators, respectively. But in lines 367-368, 'Where ξ1 and ξ2 are the weight values of safety index and efficiency index respectively.' The definitions of ξ1 and ξ2 do not match Formula 26.
7. In lines 380-382, 'And the alternative area of main insert able gap starts from 675m to 975m with a total length of 300m. Besides, ramp vehicles can use the gaps in the main lane within the alternative area to complete merging.' It is mentioned that the area of 675m-975m is used as a backup area, and ramp vehicles can use the backup area for merging. However, Figure 5 shows that only some roads in the backup area are located in the merging area, while the rest are in front of the merging area. How can this area be used for merging when the right direction is positive?
8. What do the orange and black dots in Figure 5 represent?
9. In lines 438-439, 'except 1-1, there are 7 possible split strategies for 4 vehicles, named 0-4, 1-3, 2-2, 3-1, 1-1, 1-2, 1-2, 1-2, 1-2, 1-1', the following four partition strategies are repeated.
10. The image in Figure 6 needs to be rearranged to correspond to the analysis content in the following text, and the images that have not been analyzed need to be deleted.
11. Why was the main road vehicle model for CACC constructed in the previous text, but its participation was not reflected in the subsequent experimental design?
12. The summary format of opinions and conclusions needs to be unified, such as lines 492-496, 497-500, 510-517, 541-549 and 390-396, 407-423, which have two different formats.
13. The experimental results of the article are only compared with fleets without merging strategies, and there is a lack of comparison between the PSCM method and other merging methods.
Reviewer 2 Report
This paper proposes a collaborative method for vehicle platoon. My comments are as follows:
1: The idea is interesting and the method has some potential. The authors also conducted extensive simulation experiments to demonstrate their work.
2: For the following-merge rules modeling, vehicle velocity and acceleration are important state input. However, these states could not be obtained directly. Many scholars choose to design robust estimation methods to obtain it indirectly. With new sensors configuration for autonomous vehicles, such as GNSS, IMU, and cameras, the vehicle states could be estimated precisely. Thus, some related work should be cited: autonomous vehicle kinematics and dynamics synthesis for sideslip angle estimation based on consensus kalman filter, estimation on imu yaw misalignment by fusing information of automotive onboard sensors, automated vehicle sideslip angle estimation considering signal measurement characteristic, yolov5-tassel: detecting tassels in rgb uav imagery with improved yolov5 based on transfer learning, imu-based automated vehicle body sideslip angle and attitude estimation aided by gnss using parallel adaptive kalman filters, improved vehicle localization using on-board sensors and vehicle lateral velocity, automated driving systems data acquisition and processing platform.
3: It would be better to highlight your contribution in the introduction.
Reviewer 3 Report
Page 1, Title, “A Collaborative Merging method for Connected and Automated Vehicle Platoon in Merging Area Considered Safety and Efficiency“: My suggestion is to change the title to “A Collaborative Merging method for Connected and Automated Vehicle Platoon in a Freeway Merging Area Considered Safety and Efficiency“, so to better reflect the content of the paper.
Page 1, Keywords: My suggestion is to include “freeway ramp”, “road safety and efficiency” (instead of “safety and efficiency”) and “TERCRI” in the Keywords.
Section 1. Introduction, page 1, 1st paragraph, lines 31-33: My suggestion is to include some statistics (together with their associated references) concerning traffic accidents in freeways worldwide as well as some statistics concerning traffic congestion (in terms of its cost) worldwide. I believe that these statistics will help the reader to better understand the importance of your work from the early beginning of the paper.
Section 1. Introduction, page 2, line 69: My suggestion is to add more text concerning the five references ([10-14]) since the specific topic is very interesting.
Page 4, line 161: My suggestion is to fully justify within your manuscript the selection of 80 km/h as the Vmain. Is the Vmain the same in all lanes of the freeway?
Section 2. Methods: It would be useful to include a Data Flow Chart describing all your methodological steps, if possible.
Page 11, line 347, abbreviation TERCRI: Please also include “Time Exposed Rear-end Crash Risk Index” for the benefit of the reader.
Page 13, lines 386-387: My suggestion is to fully justify within your manuscript the selection of 1500 pcu/h/lane as well as the selection of the dimensions of the simulation vehicle (4.3m long and 1.8m wide). Do these dimensions refer to a typical passenger car? Please also comment on the composition of the simulated traffic.
Page 17, line 501: My suggestion is to fully justify within your manuscript the selection of CAV proportion of 30%, 50% and 70%.
Reference list at the end of the paper: Please include the names of all the coauthors in references 5, 7, 14, 16, 17, 18, 19, 23, 26, 27, 33 and 35. Please also note that you have included both the name and the surname of the authors in some references (e.g., 9, 12, 13).
